# Recurrent Experience Replay in Distributed Reinforcement Learning

**Steven Kapturowski, Georg Ostrovski, John Quan, Rémi Munos, Will Dabney**
DeepMind, London, UK
{`skapturowski,ostrovski,johnquan,munos,wdabney`}@google.com

## Abstract

Building on the recent successes of distributed training of RL agents, in this paper we investigate the training of RNN-based RL agents from distributed prioritized experience replay. We study the effects of parameter lag resulting in representational drift and recurrent state staleness and empirically derive an improved training strategy. Using a single network architecture and fixed set of hyper-parameters, the resulting agent, Recurrent Replay Distributed DQN, quadruples the previous state of the art on Atari-57, and matches the state of the art on DMLab-30. It is the first agent to exceed human-level performance in 52 of the 57 Atari games.

## 1 Introduction

Reinforcement Learning (RL) has seen a rejuvenation of research interest recently due to repeated successes in solving challenging problems such as reaching human-level play on Atari 2600 games (Mnih et al., 2015), beating the world champion in the game of Go (Silver et al., 2017), and playing competitive 5-player DOTA (OpenAI, 2018b). The earliest of these successes leveraged experience replay for data efficiency and stacked a fixed number of consecutive frames to overcome the partial observability in Atari 2600 games. However, with progress towards increasingly difficult, partially observable domains, the need for more advanced memory-based representations increases, necessitating more principled solutions such as recurrent neural networks (RNNs). The use of LSTMs (Hochreiter & Schmidhuber, 1997) within RL has been widely adopted to overcome partial observability (Hausknecht & Stone, 2015; Mnih et al., 2016; Espeholt et al., 2018; Gruslys et al., 2018).

In this paper we investigate the training of RNNs with experience replay. We have three primary contributions. First, we demonstrate the effect of experience replay on parameter lag, leading to representational drift and recurrent state staleness. This is potentially exacerbated in the distributed training setting, and ultimately results in diminished training stability and performance. Second, we perform an empirical study into the effects of several approaches to RNN training with experience replay, mitigating the aforementioned effects. Third, we present an agent that integrates these findings to achieve significant advances in the state of the art on Atari-57 (Bellemare et al., 2013) and matches the state of the art on DMLab-30 (Beattie et al., 2016). To the best of our knowledge, our agent, Recurrent Replay Distributed DQN (R2D2), is the first to achieve this using a single network architecture and fixed set of hyper-parameters.

## 2 Background

### 2.1 Reinforcement Learning

Our work is set within the Reinforcement Learning (RL) framework (Sutton & Barto, 1998), in which an agent interacts with an environment to maximize the sum of discounted, $\gamma \in [0, 1)$, rewards. We model the environment as a Partially Observable Markov Decision Process (POMDP) given by the tuple $(\mathcal{S}, \mathcal{A}, T, R, \Omega, \mathcal{O})$ (Monahan, 1982; Jaakkola et al., 1995; Kaelbling et al., 1998). The underlying Markov Decision Process (MDP) is defined by $(\mathcal{S}, \mathcal{A}, T, R)$, where $\mathcal{S}$ is the set of states, $\mathcal{A}$ the set of actions, $T$ a transition function mapping state-actions to probability distributions over next states, and $R : \mathcal{S} \times \mathcal{A} \rightarrow \mathbb{R}$ is the reward function. Finally, $\Omega$ gives the set of observations

potentially received by the agent and $\mathcal{O}$ is the observation function mapping (unobserved) states to probability distributions over observations.

Within this framework, the agent receives an observation $o \in \Omega$, which may only contain partial information about the underlying state $s \in \mathcal{S}$. When the agent takes an action $a \in \mathcal{A}$ the environment responds by transitioning to state $s' \sim T(\cdot|s, a)$ and giving the agent a new observation $o' \sim \Omega(\cdot|s')$ and reward $r \sim R(s, a)$.

Although there are many approaches to RL in POMDPs, we focus on using recurrent neural networks (RNNs) with backpropagation through time (BPTT) (Werbos, 1990) to learn a representation that disambiguates the true state of the POMDP.

The Deep Q-Network agent (DQN) (Mnih et al., 2015) learns to play games from the Atari-57 benchmark by using frame-stacking of 4 consecutive frames as observations, and training a convolutional network to represent a value function with Q-learning (Watkins & Dayan, 1992), from data continuously collected in a replay buffer (Lin, 1993). Other algorithms like the A3C (Mnih et al., 2016), use an LSTM and are trained directly on the online stream of experience without using a replay buffer. Hausknecht & Stone (2015) combined DQN with an LSTM by storing sequences in replay and initializing the recurrent state to zero during training.

## 2.2 Distributed Reinforcement Learning

Recent advances in reinforcement learning have achieved significantly improved performance by leveraging distributed training architectures which separate learning from acting, collecting data from many actors running in parallel on separate environment instances (Horgan et al., 2018; Espeholt et al., 2018; Gruslys et al., 2018; OpenAI, 2018b;a; Jaderberg et al., 2018).

Distributed replay allows the Ape-X agent (Horgan et al., 2018) to decouple learning from acting, with actors feeding experience into the distributed replay buffer and the learner receiving (randomized) training batches from it. In addition to distributed replay with prioritized sampling (Schaul et al., 2016), Ape-X uses $n$-step return targets (Sutton, 1988), the double Q-learning algorithm (van Hasselt et al., 2016), the dueling DQN network architecture (Wang et al., 2016) and 4-frame-stacking. Ape-X achieved state-of-the-art performance on Atari-57, significantly out-performing the best single-actor algorithms. It has also been used in continuous control domains and again showed state-of-the-art results, further demonstrating the performance benefits of distributed training in RL.

IMPALA (Espeholt et al., 2018) is a distributed reinforcement learning architecture which uses a first-in-first-out queue with a novel off-policy correction algorithm called V-trace, to learn sequentially from the stream of experience generated by a large number of independent actors. IMPALA stores sequences of transitions along with an initial recurrent state in the experience queue, and since experience is trained on exactly once, this data generally stays very close to the learner parameters. Espeholt et al. (2018) showed that IMPALA could achieve strong performance in the Atari-57 and DMLab-30 benchmark suites, and furthermore was able to use a single large network to learn all tasks in a benchmark simultaneously while maintaining human-level performance.

## 2.3 The Recurrent Replay Distributed DQN Agent

We propose a new agent, the Recurrent Replay Distributed DQN (R2D2), and use it to study the interplay between recurrent state, experience replay, and distributed training. R2D2 is most similar to Ape-X, built upon prioritized distributed replay and $n$-step double Q-learning (with $n = 5$), generating experience by a large number of actors (typically 256) and learning from batches of replayed experience by a single learner. Like Ape-X, we use the dueling network architecture of Wang et al. (2016), but provide an LSTM layer after the convolutional stack, similarly to Gruslys et al. (2018).

Instead of regular $(s, a, r, s')$ transition tuples, we store fixed-length ($m = 80$) sequences of $(s, a, r)$ in replay, with adjacent sequences overlapping each other by 40 time steps, and never crossing episode boundaries. When training, we unroll both online and target networks (Mnih et al., 2015) on the same sequence of states to generate value estimates and targets. We leave details of our exact treatment of recurrent states in replay for the next sections.

Like Ape-X, we use $4$-frame-stacks and the full $18$-action set when training on Atari. On DMLab, we use single RGB frames as observations, and the same action set discretization as Hessel et al. (2018b). Following the modified Ape-X version in Pohlen et al. (2018), we do not clip rewards, but instead use an invertible value function rescaling of the form $h(x) = \text{sign}(x)(\sqrt{|x| + 1} - 1) + \epsilon x$ which results in the following $n$-step targets for the Q-value function:

$$\hat{y}_t = h \left( \sum_{k=0}^{n-1} r_{t+k} \gamma^k + \gamma^n h^{-1} \left( Q(s_{t+n}, a^*; \theta^-) \right) \right), \quad a^* = \arg\max_a Q(s_{t+n}, a; \theta).$$

Here, $\theta^-$ denotes the target network parameters which are copied from the online network parameters $\theta$ every $2500$ learner steps.

Our replay prioritization differs from that of Ape-X in that we use a mixture of max and mean absolute $n$-step TD-errors $\delta_i$ over the sequence: $p = \eta \max_i \delta_i + (1 - \eta)\bar{\delta}$. We set $\eta$ and the priority exponent to $0.9$. This more aggressive scheme is motivated by our observation that averaging over long sequences tends to wash out large errors, thereby compressing the range of priorities and limiting the ability of prioritization to pick out useful experience.

Finally, compared to Ape-X, we used the slightly higher discount of $\gamma = 0.997$, and disabled the loss-of-life-as-episode-end heuristic that has been used in Atari agents in some of the work since (Mnih et al., 2015). A full list of hyper-parameters is provided in the Appendix.

We train the R2D2 agent with a single GPU-based learner, performing approximately 5 network updates per second (each update on a mini-batch of $64$ length-$80$ sequences), and each actor performing $\sim 260$ environment steps per second on Atari ($\sim 130$ per second on DMLab).

## 3 TRAINING RECURRENT RL AGENTS WITH EXPERIENCE REPLAY

In order to achieve good performance in a partially observed environment, an RL agent requires a state representation that encodes information about its state-action trajectory in addition to its current observation. The most common way to achieve this is by using an RNN, typically an LSTM (Hochreiter & Schmidhuber, 1997), as part of the agent's state encoding. To train an RNN from replay and enable it to learn meaningful long-term dependencies, whole state-action trajectories need to be stored in replay and used for training the network. Hausknecht & Stone (2015) compared two strategies of training an LSTM from replayed experience:

- Using a zero start state to initialize the network at the beginning of sampled sequences.
- Replaying whole episode trajectories.

The zero start state strategy's appeal lies in its simplicity, and it allows independent decorrelated sampling of relatively short sequences, which is important for robust optimization of a neural network. On the other hand, it forces the RNN to learn to recover meaningful predictions from an atypical initial recurrent state ('initial recurrent state mismatch'), which may limit its ability to fully rely on its recurrent state and learn to exploit long temporal correlations. The second strategy on the other hand avoids the problem of finding a suitable initial state, but creates a number of practical, computational, and algorithmic issues due to varying and potentially environment-dependent sequence length, and higher variance of network updates because of the highly correlated nature of states in a trajectory when compared to training on randomly sampled batches of experience tuples.

Hausknecht & Stone (2015) observed little difference between the two strategies for empirical agent performance on a set of Atari games, and therefore opted for the simpler zero start state strategy. One possible explanation for this is that in some cases, an RNN tends to converge to a more 'typical' state if allowed a certain number of 'burn-in' steps, and so recovers from a bad initial recurrent state on a sufficiently long sequence. We also hypothesize that while the zero start state strategy may suffice in the mostly fully observable Atari domain, it prevents a recurrent network from learning actual long-term dependencies in more memory-critical domains (e.g. on DMLab).

To fix these issues, we propose and evaluate two strategies for training a recurrent neural network from randomly sampled replay sequences, that can be used individually or in combination:

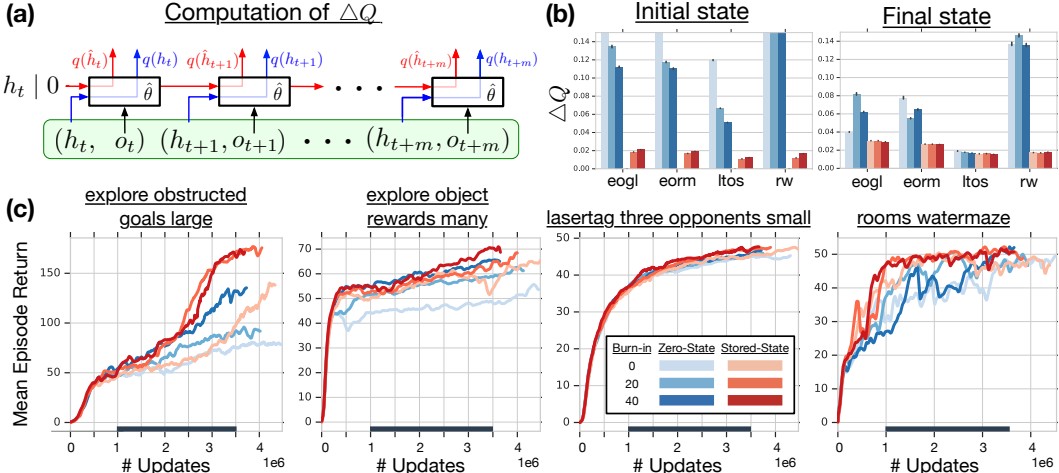

Figure 1: Top row shows Q-value discrepancy $\Delta Q$ as a measure for recurrent state staleness. **(a)** Diagram of how $\Delta Q$ is computed, with green box indicating a whole sequence sampled from replay. For simplicity, $l = 0$ (no burn-in). **(b)** $\Delta Q$ measured at first state and last state of replay sequences, for agents training on a selection of DMLab levels (indicated by initials) with different training strategies. Bars are averages over seeds and through time indicated by bold line on x-axis in bottom row. **(c)** Learning curves on the same levels, varying the training strategy, and averaged over 3 seeds.

- **Stored state:** *Storing the recurrent state in replay and using it to initialize the network at training time.* This partially remedies the weakness of the zero start state strategy, however it may suffer from the effect of 'representational drift' leading to 'recurrent state staleness', as the stored recurrent state generated by a sufficiently old network could differ significantly from a typical state produced by a more recent version.

- **Burn-in:** *Allow the network a 'burn-in period' by using a portion of the replay sequence only for unrolling the network and producing a start state, and update the network only on the remaining part of the sequence.* We hypothesize that this allows the network to partially recover from a poor start state (zero, or stored but stale) and find itself in a better initial state before being required to produce accurate outputs.

In all our experiments we will be using the proposed agent architecture from Section 2.3 with replay sequences of length $m = 80$, with an optional burn-in prefix of $l = 40$ or 20 steps. Our aim is to assess the negative effects of representational drift and recurrent state staleness on network training and how they are mitigated by the different training strategies. For that, we will compare the Q-values produced by the network on sampled replay sequences when unrolled using one of these strategies and the Q-values produced when using the true stored recurrent states at each step (see Figure 1a, showing different sources for the hidden state).

More formally, let $o_t, \ldots, o_{t+m}$ and $h_t, \ldots, h_{t+m}$ denote the replay sequence of observations and stored recurrent states, and denote by $h_{t+1} = h(o_t, h_t; \theta)$ and $q(h_t; \theta)$ the recurrent state and Q-value vector output by the recurrent neural network with parameter vector $\theta$, respectively. We write $\hat{h}_t$ for the hidden state, used during training and initialized under one of the above strategies (either $\hat{h}_t = 0$ or $\hat{h}_t = h_t$). Then $\hat{h}_{t+i} = h(o_{t+i-1}, \hat{h}_{t+i-1}; \hat{\theta})$ is computed by unrolling the network with parameters $\hat{\theta}$ on the sequence $o_t, \ldots, o_{t+l+m-1}$. We estimate the impact of representational drift and recurrent state staleness by their effect on the Q-value estimates, by measuring *Q-value discrepancy*

$$\Delta Q = \frac{\|q(\hat{h}_{t+i}; \hat{\theta}) - q(h_{t+i}; \hat{\theta})\|_2}{|\max_{a,j}(q(\hat{h}_{t+j}; \hat{\theta}))_a|}$$

for the first ($i = l$) and last ($i = l + m - 1$) states of the non-burn-in part of the replay sequence (see Figure 1a for an illustration). The normalization by the maximal Q-value helps comparability between different environments and training stages, as the Q-value range of an agent can vary dras-

tically between these. Note that we are *not* directly comparing the Q-values produced at acting and training time, $q(h_t; \theta)$ and $q(\hat{h}_t; \hat{\theta})$, as these can naturally be expected to be distinct as the agent is being trained. Instead we focus on the difference that results from applying the same network (parameterized by $\hat{\theta}$) to the distinct recurrent states.

In Figure 1b, we are comparing agents trained with the different strategies on several DMLab environments in terms of this proposed metric. It can be seen that the zero start state heuristic results in a significantly more severe effect of recurrent state staleness on the outputs of the network. As hypothesized above, this effect is greatly reduced for the last sequence states compared to the first ones, after the RNN has had time to recover from the atypical start state, but the effect of staleness is still substantially worse here for the zero state than the stored state strategy. Another potential downside of the pure zero state heuristic is that it prevents the agent from strongly relying on its recurrent state and exploit long-term temporal dependencies, see Section 5.

We observe that the burn-in strategy on its own partially mitigates the staleness problem on the initial part of replayed sequences, while not showing a significant effect on the Q-value discrepancy for later sequence states. Empirically, this translates into noticeable performance improvements, as can be seen in Figure 1c. This itself is noteworthy, as the only difference between the pure zero state and the burn-in strategy lies in the fact that the latter unrolls the network over a prefix of states on which the network does not receive updates. In informal experiments (not shown here) we observed that this is not due to the different unroll lengths themselves (i.e., the zero state strategy without burn-in, on sequences of length $l + m$, performed worse overall). We hypothesize that the beneficial effect of burn-in lies in the fact that it prevents 'destructive updates' to the RNN parameters resulting from highly inaccurate initial outputs on the first few time steps after a zero state initialization.

The stored state strategy, on the other hand, proves to be overall much more effective at mitigating state staleness in terms of the Q-value discrepancy, which also leads to clearer and more consistent improvements in empirical performance. Finally, the combination of both methods consistently yields the smallest discrepancy on the last sequence states and the most robust performance gains.

We conclude the section with the observation that both stored state and burn-in strategy provide substantial advantages over the naive zero state training strategy, in terms of (indirect) measures of the effect of representation drift and recurrent state staleness, and empirical performance. Since they combine beneficially, we use both of these strategies (with burn-in length of $l = 40$) in the empirical evaluation of our proposed agent in Section 4. Additional results on the effects of distributed training on representation drift and Q-value discrepancy are given in the Appendix.

## 4 EXPERIMENTAL EVALUATION

In this section we evaluate the empirical performance of R2D2 on two challenging benchmark suites for deep reinforcement learning: Atari-57 (Bellemare et al., 2013) and DMLab-30 (Beattie et al., 2016). One of the fundamental contributions of Deep Q-Networks (DQN) (Mnih et al., 2015) was to set as standard practice the use of a single network architecture and set of hyper-parameters across the entire suite of 57 Atari games. Unfortunately, expanding past Atari this standard has not been maintained and, to the best of our knowledge, at present there is no algorithm applied to both Atari-57 and DMLab-30 under this standard. In particular, we will compare performance with Ape-X and IMPALA for which hyper-parameters are tuned separately for each benchmark.

For R2D2, we use a single neural network architecture and a single set of hyper-parameters across all experiments. This demonstrates greater robustness and generality than has been previously observed in deep RL. It is also in pursuit of this generality, that we decided to disable the (Atari-specific) heuristic of treating life losses as episode ends, and did not apply reward clipping. Despite this, we observe state-of-the-art performance in both Atari and DMLab, validating the intuitions derived from our empirical study. A more detailed ablation study of the effects of these modifications is presented in the Appendix.

### 4.1 ATARI-57

The Atari-57 benchmark is built upon the Arcade Learning Environment (ALE) (Bellemare et al., 2013), and consists of 57 classic Atari 2600 video games. Initial human-level performance was

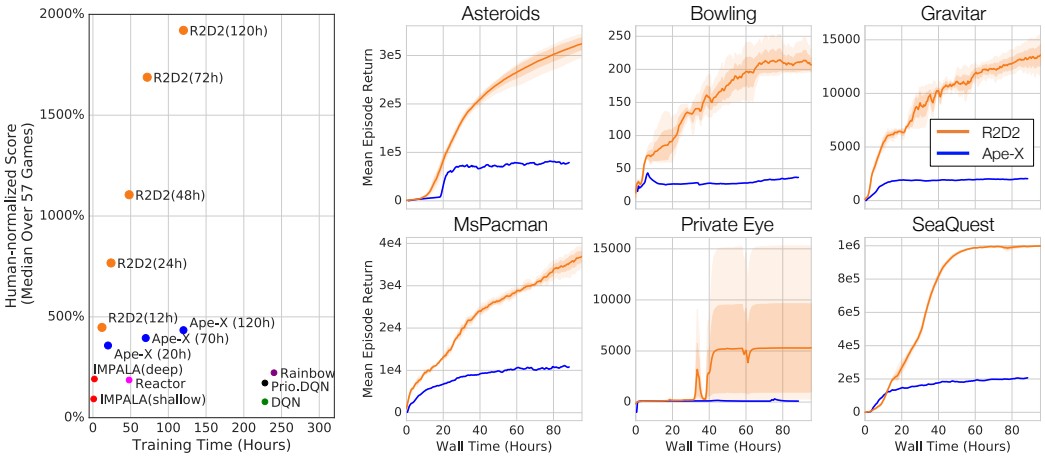

Figure 2: Atari-57 results. **Left:** median human-normalized scores and training times of various agent architectures. Diagram reproduced and extended from (Horgan et al., 2018). **Right:** Example individual learning curves of R2D2, averaged over 3 seeds, and Ape-X, single seed.

achieved by DQN (Mnih et al., 2015), and since then RL agents have improved significantly through both algorithmic and architectural advances. Currently, state of the art for a single actor is achieved by the recent distributional reinforcement learning algorithms IQN (Dabney et al., 2018) and Rainbow (Hessel et al., 2018a), and for multi-actor results, Ape-X (Horgan et al., 2018).

Figure 2 (left) shows the median human-normalized scores across all games for R2D2 and related methods (see Appendix for full Atari-57 scores and learning curves). R2D2 achieves an order of magnitude higher performance than all single-actor agents and quadruples the previous state-of-the-art performance of Ape-X using fewer actors (256 instead of 360), resulting in higher sample- and time-efficiency. Table 1 lists mean and median human-normalized scores for R2D2 and other algorithms, highlighting these improvements.

In addition to achieving state-of-the-art results on the entire task suite, R2D2 also achieves the highest ever reported agent scores on a large fraction of the individual Atari games, in many cases 'solving' the respective games by achieving the highest attainable score. In Figure 2 (right) we highlight some of these individual learning curves of R2D2. As an example, notice the performance on MS.PACMAN is even greater than that of the agent reported in (van Seijen et al., 2017), which was engineered specifically for this game. Furthermore, we notice that Ape-X achieves super-human performance for the same number of games as Rainbow (49), and that its improvements came from improving already strong scores. R2D2 on the other hand is super-human on 52 out of 57 games. Of those remaining, we anecdotally observed that three (SKIING, SOLARIS, and PRIVATE EYE) can reach super-human performance with higher discount rates and faster target network updates. The other two (MONTEZUMA'S REVENGE and PITFALL) are known hard exploration problems, and solving these with a general-purpose algorithm will likely require new algorithmic insights.

## 4.2 DMLAB-30

DMLab-30 is a suite of 30 problems set in a 3D first-person game engine, testing for a wide range of different challenges (Beattie et al., 2016). While Atari can largely be approached with only frame-stacking, DMLab-30 requires long-term memory to achieve reasonable performance. Perhaps because of this, and the difficulty of integrating recurrent state with experience replay, top-performing agents have, to date, always come in the form of actor-critic algorithms trained in (near) on-policy settings. For the first time we show state-of-the-art performance on DMLab-30 using a value-function-based agent. We stress that, different from the state-of-the-art IMPALA architecture (Espeholt et al., 2018), the R2D2 agent uses the same set of hyper-parameters here as on Atari.

Here we are mainly interested in comparing to the IMPALA 'experts', not to its multi-task variant. Since the original IMPALA experts were trained on a smaller amount of data (approximately 333M

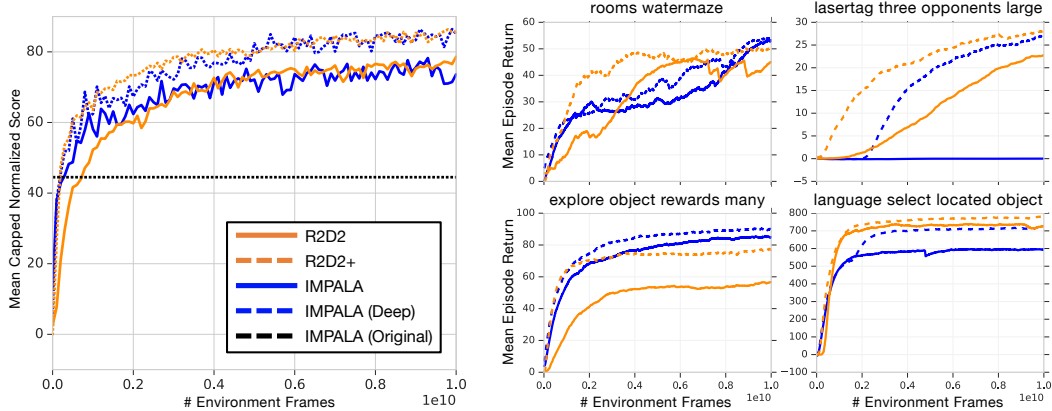

Figure 3: DMLab-30 comparison of R2D2 and R2D2+ with our re-run of IMPALA shallow and deep in terms of mean-capped human-normalized score (Espeholt et al., 2018).

| Human-Normalized Score | Atari-57 | | DMLab-30 | |
|---|---|---|---|---|
| | **Median** | **Mean** | **Median** | **Mean-Capped** |
| Ape-X (Horgan et al., 2018) | 434.1% | 1695.6% | – | – |
| Reactor (Gruslys et al., 2018) | 187.0% | – | – | – |
| IMPALA, deep (Espeholt et al., 2018) | 191.8% | 957.6% | 49.0% | 45.8% |
| IMPALA, shallow (re-run) | – | – | 89.7% | 73.6% |
| IMPALA, deep (re-run) | – | – | **107.5%** | 85.1% |
| R2D2+ | – | – | 99.5% | **85.7%** |
| R2D2, feed-forward | 589.2% | 1974.4% | – | – |
| R2D2 | **1920.6%** | **4024.9%** | 96.9% | 78.3% |

Table 1: Comparison of Atari-57 and DMLab-30 results. R2D2 average *final* score over 3 seeds (1 seed for feed-forward variant), IMPALA *final* score over 1 seed, Ape-X *best* training score with 1 seed. Our re-run of IMPALA uses the same improved action set from (Hessel et al., 2018b) as R2D2, and is trained for a comparable number of environment frames (10B frames; the original IMPALA experts in (Espeholt et al., 2018) were only trained for approximately 333M frames). R2D2+ refers to the adapted R2D2 variant matching deep IMPALA's 15-layer ResNet architecture and asymmetric reward clipping, as well as using a shorter target update period of 400.

environment frames) and since R2D2 uses the improved action set introduced in (Hessel et al., 2018b), we decided to re-run the IMPALA agent with improved action set and for a comparable training time (10B environment frames) for a fairer comparison, resulting in substantially improved scores for the IMPALA agent compared to the original in (Espeholt et al., 2018), see Table 1.

Figure 3 compares R2D2 with IMPALA. We note that R2D2 exceeds the performance of the (shallow) IMPALA version, despite using the exact same set of hyper-parameters and architecture as the variant trained on Atari, and in particular not using the 'optimistic asymmetric reward clipping' used by all IMPALA agents[1].

To demonstrate the potential of our agent, we also devise a somewhat adapted R2D2 version for DM-Lab only (R2D2+) by adding asymmetric reward clipping, using the 15-layer ResNet from IMPALA (deep), and reducing the target update frequency from 2500 to 400 for better sample efficiency. To fit the larger model in GPU memory, we reduced the batch size from 64 to 32 in these runs only. We observe that this modified version yields further substantial improvements over standard R2D2 and matches deep IMPALA in terms of sample efficiency as well as asymptotic performance. Both our re-run of deep IMPALA and R2D2+ are setting new state-of-the-art scores on the DMLab-30 benchmark.

---

[1]This domain-specific heuristic aids exploration on a certain subset of the DMLab-30 levels.

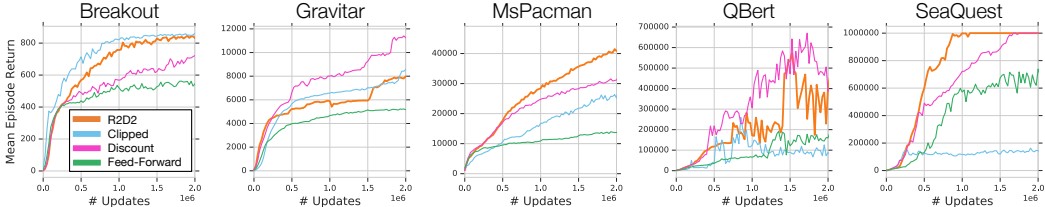

Figure 4: Ablations with reward clipping instead of value function rescaling (**Clipped**), smaller discount factor of $\gamma = 0.99$ (**Discount**), and feed-forward (**Feed-Forward**) variants of R2D2.

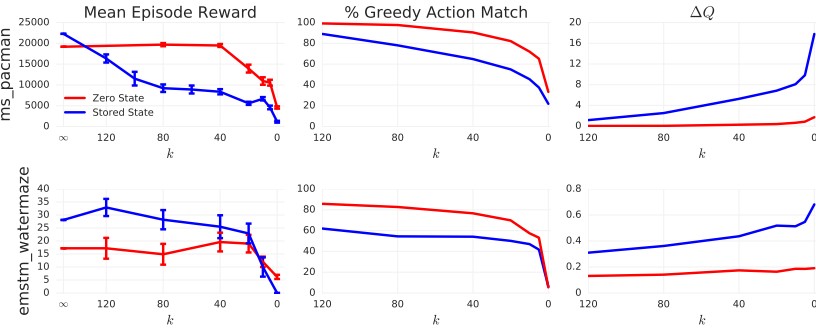

Figure 5: Effect of restricting R2D2's policy's memory on MS.PACMAN and EMSTM_WATERMAZE.

## 5 ANALYSIS OF AGENT PERFORMANCE

Atari-57 is a class of environments which are almost fully observable (given 4-frame-stack observations), and agents trained on it are not necessarily expected to strongly benefit from a memory-augmented representation. The main algorithmic difference between R2D2 and its predecessor, Ape-X, is the use of a recurrent neural network, and it is therefore surprising by how large a margin R2D2 surpasses the previous state of the art on Atari. In this section we analyze the role of the LSTM network and other algorithmic choices for the high performance of the R2D2 agent.

Since the performance of asynchronous or distributed RL agents can depend on subtle implementational details and even factors such as precise hardware setup, it is impractical to perform a direct comparison to the Ape-X agent as reported in (Horgan et al., 2018). Instead, here we verify that the LSTM and its training strategy play a crucial role for the success of R2D2 by a comparison of the R2D2 agent with a purely feed-forward variant, all other parameters held fixed. Similarly, we consider the performance of R2D2 using reward clipping without value rescaling (Clipped) and using a smaller discount factor of $\gamma = 0.99$ (Discount). The ablation results in Figure 4 show very clearly that the LSTM component is crucial for boosting the agent's peak performance as well as learning speed, explaining much of the performance difference to Ape-X. Other design choices have more mixed effects, improving in some games and hurting performance in others. Full ablation results (in particular, an ablation over the full Atari-57 suite of the feed-forward agent variant, as well as an ablation of the use of the life-loss-as-episode-termination heuristic) are presented in the Appendix.

In our next experiment we test to what extent the R2D2 agent relies on its memory, and how this is impacted by the different training strategies. For this we select the Atari game MS.PACMAN, on which R2D2 shows state-of-the-art performance despite the game being virtually fully observable, and the DMLab task EMSTM_WATERMAZE, which strongly requires the use of memory. We train two agents on each game, using the zero and stored state strategies, respectively. We then evaluate these agents by restricting their policy to a fixed history length: at time step $t$, their policy uses an LSTM unrolled over time steps $o_{t-k+1}, \ldots, o_t$, with the hidden state $h_{t-k}$ replaced by zero instead of the actual hidden state (note this is only done for evaluation, not at training time of the agents).

In Figure 5 (left) we decrease the history length $k$ from $\infty$ (full history) down to $0$ and show the degradation of agent performance (measured as mean score over 10 episodes) as a function of $k$. We additionally show the difference of max-Q-values and the percentage of correct greedy actions (where the unconstrained variant is taken as ground truth).

We observe that restricting the agent's memory gradually decreases its performance, indicating its nontrivial use of memory on both domains. Crucially, while the agent trained with stored state shows higher performance when using the full history, its performance decays much more rapidly than for the agent trained with zero start states. This is evidence that the zero start state strategy, used in past RNN-based agents with replay, limits the agent's ability to learn to make use of its memory. While this doesn't necessarily translate into a performance difference (like in MS.PACMAN), it does so whenever the task requires an effective use of memory (like EMSTM_WATERMAZE). This advantage of the stored state compared to the zero state strategy may explain the large performance difference between R2D2 and its close cousin Reactor (Gruslys et al., 2018), which trains its LSTM policy from replay with the zero state strategy.

Finally, the right and middle columns of Figure 5 show a monotonic decrease of the quality of Q-values and the resulting greedy policy as the available history length $k$ is decreased to $0$, providing a simple causal link between the constraint and the empirical agent performance.

For a qualitative comparison of different behaviours learned by R2D2 and its feed-forward variant, we provide several agent videos at `https://bit.ly/r2d2600`.

# 6    CONCLUSIONS

Here we take a step back from evaluating performance and discuss our empirical findings in a broader context. There are two surprising findings in our results.

First, although zero state initialization was often used in previous works (Hausknecht & Stone, 2015; Gruslys et al., 2018), we have found that it leads to misestimated action-values, especially in the early states of replayed sequences. Moreover, without burn-in, updates through BPTT to these early time steps with poorly estimated outputs seem to give rise to destructive updates and hinder the network's ability to recover from sub-optimal initial recurrent states. This suggests that either the context-dependent recurrent state should be stored along with the trajectory in replay, or an initial part of replayed sequences should be reserved for burn-in, to allow the RNN to rely on its recurrent state and exploit long-term temporal dependencies, and the two techniques can also be combined beneficially. We have also observed that the underlying problems of representational drift and recurrent state staleness are potentially exacerbated in the distributed setting (see Appendix), highlighting the importance of robustness to these effects through an adequate training strategy of the RNN.

Second, we found that the impact of RNN training goes beyond providing the agent with memory. Instead, RNN training also serves a role not previously studied in RL, potentially by enabling better representation learning, and thereby improves performance even on domains that are fully observable and do not obviously require memory (cf. BREAKOUT results in the feed-forward ablation).

Finally, taking a broader view on our empirical results, we note that scaling up of RL agents through parallelization and distributed training allows them to benefit from huge experience throughput and achieve ever-increasing results over broad simulated task suites such as Atari-57 and DMLab-30. Impressive as these results are in terms of raw performance, they come at the price of high sample complexity, consuming billions of simulated time steps in hours or days of wall-clock time. One widely open avenue for future work lies in improving the sample efficiency of these agents, to allow applications to domains that do not easily allow fast simulation at similar scales. Another remaining challenge, very apparent in our results on Atari-57, is exploration: Save for the hardest-exploration games from Atari-57, R2D2 surpasses human-level performance on this task suite significantly, essentially 'solving' many of the games therein.

ACKNOWLEDGMENTS

We are tremendously grateful to our colleagues at DeepMind and Google for their contributions to this work, as well as to the anonymous reviewers for their feedback. We thank Hado van Hasselt, Tom Schaul, and David Silver for feedback on an earlier draft of the paper. We thank Toby Pohlen, Bilal Piot, Tom Stepleton, and Mohammad Gheshlaghi Azar for helpful conversations and advice during development and writing.

We want to thank John Aslanides for many engineering contributions to the underlying agent infrastructure, and Gabriel Barth-Maron, Manuel Kroiss, David Budden, Jackie Kay, Matt Hoffman for library and training infrastructure support.

Sean Silva provided an incredibly comprehensive analysis of our initial submission results that made several novel observations about the benefits of memory in Atari. Finally, for generously providing raw data of previous agents, we wish to thank Hubert Soyer, Dan Horgan, Matteo Hessel, Simon Schmitt, Lasse Espeholt, and Audrunas Gruslys.

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

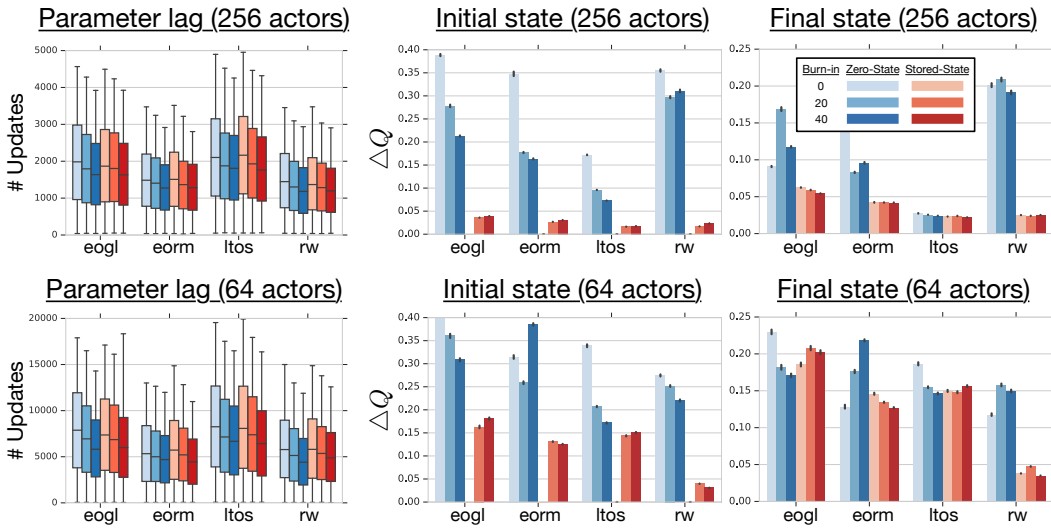

Figure 6: **Left:** Parameter lag experienced with distributed prioritized replay with (top) 256 and (bottom) 64 actors on four DMLab levels: explore obstructed goals large (eogl), explore object rewards many (eorm), lasertag three opponents small (lots), rooms watermaze (rw). **Center:** initial-state and **Right:** final-state Q-value discrepancy for the same set of experiments.

## APPENDIX

### EFFECT OF DISTRIBUTED RL AGENT TRAINING

In this section, we investigate the effects of distributed training of an agent using a recurrent neural network, where a large number of actors feed their experience into a replay buffer for a single learner.

On the one hand, the distributed setting typically presents a less severe problem of representational drift than the single-actor case, such as the one studied in (Hausknecht & Stone, 2015). This is because in relative terms, the large amount of generated experience is replayed less frequently (on average, an experience sample is replayed less than once in the Ape-X agent, compared to eight times in DQN), and so distributed agent training tends to give rise to a smaller degree of 'parameter lag' (the mean age, in parameter updates, of the network parameters used to generate an experience, at the time it is being replayed).

On the other hand, the distributed setting allows for easy scaling of computational resources according to hardware or time constraints. An ideal distributed agent should therefore be robust to changes in, e.g., the number of actors, without careful parameter re-tuning. As we have seen in the previous section, RNN training from replay is sensitive to the issue of representational drift, the severity of which can depend on exactly these parameters.

To investigate these effects, we train the R2D2 agent with a substantially smaller number of actors. This has a direct (inversely proportional) effect on the parameter lag (see Figure 6(left)). Specifically, in our experiments, as the number of actors is changed from $256$ to $64$, the mean parameter lag goes from $1500$ to approximately $5500$ parameter updates, which in turn impacts the magnitude of representation drift and recurrent state staleness, as measured by $\Delta Q$ in Section 3.

The right two columns in Figure 6 show an overall increase of the average $\Delta Q$ for the smaller number of actors, both for first and last states of replayed sequences. This supports the above intuitions and highlights the increased importance of an improved training strategy (compared to the zero state strategy) in the distributed training setting, if a certain level of empirical agent performance is to be maintained across ranges of extrinsic and potentially hardware dependent parameters.

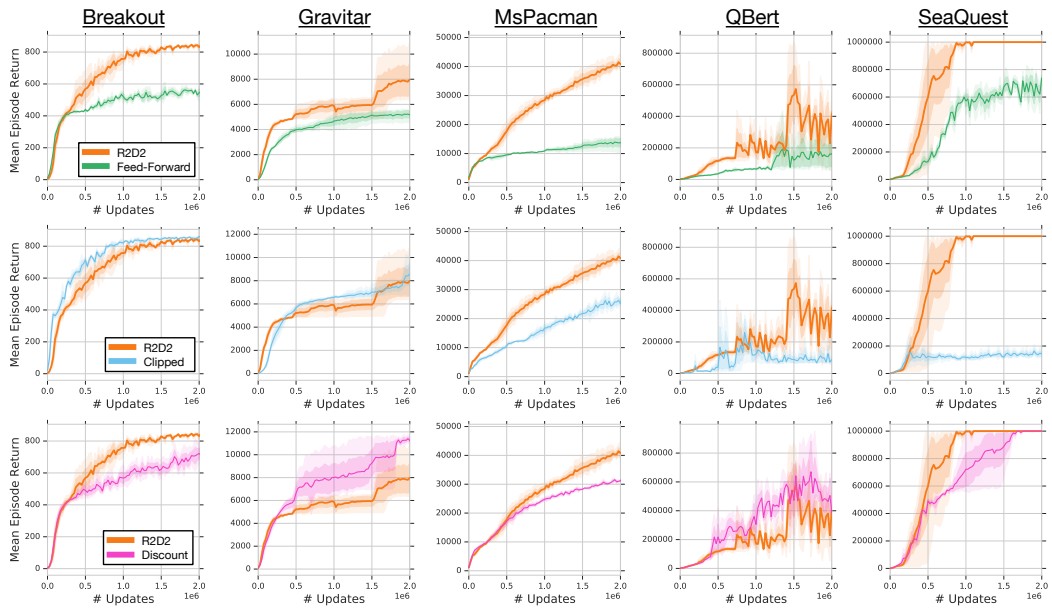

Figure 7: Ablation results with standard deviations shown by shading (3 seeds). 'Clipped' refers to the agent variant using clipped rewards (instead of value function rescaling), 'discount' refers to the use of a discount value of 0.99 (instead of 0.997).

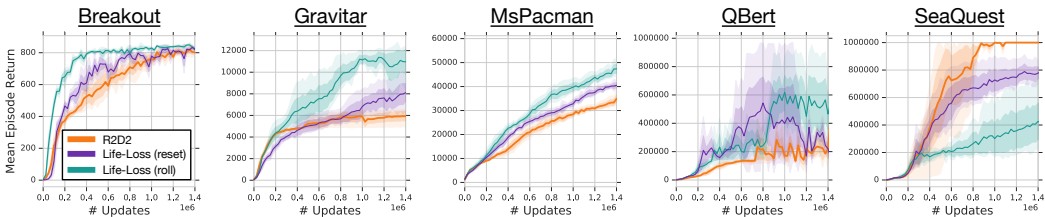

Figure 8: Ablation results for the use of life loss as episode termination on Atari, with standard deviations shown by shading (3 seeds). 'reset' refers to the agent variant using life losses as full episode terminations (preventing value function bootstrapping across life loss events, as well as resetting the LSTM state), whereas 'roll' only prevents value function bootstrapping, but unrolls the LSTM for the duration of a full episode (potentially spanning multiple life losses).

EXTENDED RESULTS

In this section we give additional experimental results supporting our empirical study in the main text. Figure 7 gives a more in-depth view of the ablation results from Figure 4. We see that, with the exception of the feed-forward ablation, there are always games in which the ablated choice performs better. Our choice of architecture and configuration optimizes for overall performance and general (cross-domain) applicability, but for individual games there are different configurations that would yield improved performance.

Additionally, in Figure 8 we compare R2D2 with variants using the life loss signal as episode termination. Both ablation variants interrupt value function bootstrapping past the life loss events, but differ in that one ('reset') also resets the LSTM state at these events, whereas the other ('roll') only resets the LSTM state at actual episode boundaries, like regular R2D2. Despite the fact that the life loss heuristic is generally helpful to speed up learning in Atari, we did not use it in our main R2D2 agent for the sake of generality of the algorithm.

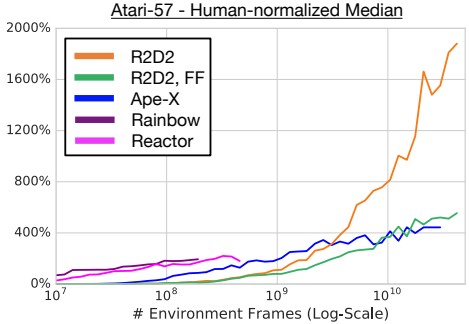

Figure 9: Comparing sample efficiency between state-of-the-art agents on Atari-57. We observe a general trend of increasing final performance being negatively correlated with sample efficiency, which holds for all four algorithms compared.

In Figure 9 we compare the sample efficiency of R2D2 with recent state-of-the-art agents on Atari-57 in terms of human-normalized median score. As expected, the more distributed agents have worse sample efficiency early on, but also much improved long-term performance. This is an interesting correlation on its own, but we add that R2D2 appears to achieve a qualitatively different performance curve than any of the other algorithms.

Note that, while Ape-X has a larger number of actors than R2D2 (360 compared to 256), its learner processes approximately 20 batches of size 512 per second, whereas R2D2 performs updates on batches of $64 \times 80$ observations (batch size $\times$ sequence length), at a rate of approximately 5 per second. This results in a reduced 'replay ratio' (effective number of times each experienced observation is being replayed): On average, Ape-X replays each observation approximately 1.3 times, whereas this number is only about 0.8 for R2D2, which explains the initial sample efficiency advantage of Ape-X.

HYPER-PARAMETERS

R2D2 uses the same 3-layer convolutional network as DQN (Mnih et al., 2015), followed by an LSTM with 512 hidden units, which feeds into the advantage and value heads of a dueling network (Wang et al., 2016), each with a hidden layer of size 512. Additionally, the LSTM receives as input the reward and one-hot action vector from the previous time step. On the four language tasks in the DMLab suite, we are using the same additional language-LSTM with 64 hidden units as IMPALA (Espeholt et al., 2018).

| | |
|---|---|
| Number of actors | 256 |
| Actor parameter update interval | 400 environment steps |
| Sequence length $m$ | 80 (+ prefix of $l = 40$ in burn-in experiments) |
| Replay buffer size | $4 \times 10^6$ observations ($10^5$ part-overlapping sequences) |
| Priority exponent | 0.9 |
| Importance sampling exponent | 0.6 |
| Discount $\gamma$ | 0.997 |
| Minibatch size | 64 (32 for R2D2+ on DMLab) |
| Optimizer | Adam (Kingma & Ba, 2014) |
| Optimizer settings | learning rate $= 10^{-4}$, $\varepsilon = 10^{-3}$ |
| Target network update interval | 2500 updates |
| Value function rescaling | $h(x) = \text{sign}(x)(\sqrt{|x| + 1} - 1) + \epsilon x$, $\epsilon = 10^{-3}$ |

Table 2: Hyper-parameters values used in R2D2. All missing parameters follow the ones in Ape-X (Horgan et al., 2018).

As is usual for agent training on Atari since (Mnih et al., 2015), we cap all (training and evaluation) episodes at 30 minutes ($108,000$ environment frames). All reported scores are (undiscounted)

episode returns. For the R2D2 agent we always report *final* scores (as opposed to some of the comparison agents that use *best* score attained during training).

FULL RESULTS

| GAMES | IMPALA(s) | IMPALA(d) | R2D2 | R2D2+ |
|---|---|---|---|---|
| rooms_collect_good_objects_test | **10.0** | **10.0** | 9.8 | 9.9 |
| rooms_exploit_deferred_effects_test | 46.0 | **62.2** | 39.9 | 38.1 |
| rooms_select_nonmatching_object | 11.0 | 39.0 | 2.3 | **63.6** |
| rooms_watermaze | 48.4 | 47.0 | 45.9 | **49.0** |
| rooms_keys_doors_puzzle | 43.8 | **54.6** | 52.0 | 46.2 |
| language_select_described_object | 638.0 | **664.0** | 445.9 | 617.6 |
| language_select_located_object | 534.8 | 731.4 | 555.4 | **772.8** |
| language_execute_random_task | **515.2** | 465.4 | 363.1 | 497.4 |
| language_answer_quantitative_question | 18.0 | **362.0** | 258.0 | 344.4 |
| lasertag_one_opponent_large | **0.0** | **0.0** | **0.0** | **0.0** |
| lasertag_three_opponents_large | 0.0 | **32.2** | 22.9 | 28.6 |
| lasertag_one_opponent_small | 0.0 | 0.0 | 0.0 | **31.8** |
| lasertag_three_opponents_small | 53.8 | **57.2** | 42.8 | 49.0 |
| natlab_fixed_large_map | 57.0 | **63.4** | 48.9 | 60.6 |
| natlab_varying_map_regrowth | 22.4 | **34.0** | 25.7 | 24.6 |
| natlab_varying_map_randomized | **48.6** | 47.0 | 40.8 | 42.4 |
| skymaze_irreversible_path_hard | 80.0 | 80.0 | **84.0** | 76.0 |
| skymaze_irreversible_path_varied | 80.0 | **100.0** | 80.0 | 76.0 |
| psychlab_arbitrary_visuomotor_mapping | 37.0 | **38.4** | 34.2 | 33.1 |
| psychlab_continuous_recognition | **30.2** | 28.6 | 29.4 | 30.0 |
| psychlab_sequential_comparison | **30.8** | 29.6 | **30.8** | 30.0 |
| psychlab_visual_search | **80.0** | **80.0** | 79.6 | 79.9 |
| explore_object_locations_small | **104.0** | 100.4 | 82.9 | 83.7 |
| explore_object_locations_large | 78.8 | **91.0** | 52.4 | 60.6 |
| explore_obstructed_goals_small | 324.0 | **372.0** | 357.6 | 311.9 |
| explore_obstructed_goals_large | **150.0** | 102.0 | 63.2 | 95.5 |
| explore_goal_locations_small | **566.0** | 482.0 | 459.6 | 460.7 |
| explore_goal_locations_large | **386.0** | 316.0 | 328.0 | 174.7 |
| explore_object_rewards_few | 39.0 | **92.6** | 76.5 | 80.7 |
| explore_object_rewards_many | 79.8 | **89.4** | 59.7 | 75.8 |

Table 3: Performance of R2D2 and R2D2+, averaged over 3 seeds, compared to our own single-seed re-run of IMPALA (shallow/deep) with improved action-set and trained on the same amount of data (10B environment frames). Compared to standard R2D2, the R2D2+ variant uses a shorter target network update frequency (400 compared to 2500), as well as the substantially larger 15-layer ResNet and the custom 'optimistic asymmetric reward clipping' from (Espeholt et al., 2018).

| GAMES | HUMAN | REACTOR | IMPALA(S/D) | APE-X | R2D2 |
|---|---|---|---|---|---|
| alien | 7127.8 | 6482.1 | 1536.0/15962.1 | 40804.9 | **229496.9** |
| amidar | 1719.5 | 833.0 | 497.6/1554.8 | 8659.2 | **29321.4** |
| assault | 742.0 | 11013.5 | 12086.9/19148.5 | 24559.4 | **108197.0** |
| asterix | 8503.3 | 36238.5 | 29692.5/300732.0 | 313305.0 | **999153.3** |
| asteroids | 47388.7 | 2780.3 | 3508.1/108590.1 | 155495.1 | **357867.7** |
| atlantis | 29028.1 | 308257.5 | 773355.5/849967.5 | 944497.5 | **1620764.0** |
| bank_heist | 753.1 | 988.7 | 1200.3/1223.2 | 1716.4 | **24235.9** |
| battle_zone | 37187.5 | 61220.0 | 13015.0/20885.0 | 98895.0 | **751880.0** |
| beam_rider | 16926.5 | 8566.5 | 8219.9/32463.5 | 63305.2 | **188257.4** |
| berzerk | 2630.4 | 1641.4 | 888.3/1852.7 | **57196.7** | 53318.7 |
| bowling | 160.7 | 75.4 | 35.7/59.9 | 17.6 | **219.5** |
| boxing | 12.1 | 99.4 | 96.3/**100.0** | **100.0** | 98.5 |
| breakout | 30.5 | 518.4 | 640.4/787.3 | 800.9 | **837.7** |
| centipede | 12017.0 | 3402.8 | 5528.1/11049.8 | 12974.0 | **599140.3** |
| chopper_command | 7387.8 | 37568.0 | 5012.0/28255.0 | 721851.0 | **986652.0** |
| crazy_climber | 35829.4 | 194347.0 | 136211.5/136950.0 | 320426.0 | **366690.7** |
| defender | 18688.9 | 113127.8 | 58718.3/185203.0 | 411943.5 | **665792.0** |
| demon_attack | 1971.0 | 100188.5 | 107264.7/132827.0 | 133086.4 | **140002.3** |
| double_dunk | -16.4 | 11.4 | -0.4/-0.3 | 12.8 | **23.7** |
| enduro | 860.5 | 2230.1 | 0.0/0.0 | 2177.4 | **2372.7** |
| fishing_derby | -38.8 | 23.2 | 32.1/44.9 | 44.4 | **85.8** |
| freeway | 29.6 | 31.4 | 0.0/0.0 | **33.7** | 32.5 |
| frostbite | 4334.7 | 8042.1 | 269.6/317.8 | 9328.6 | **315456.4** |
| gopher | 2412.5 | 69135.1 | 1002.4/66782.3 | 120500.9 | **124776.3** |
| gravitar | 3351.4 | 1073.8 | 211.5/359.5 | 1598.5 | **15680.7** |
| hero | 30826.4 | 35542.2 | 33853.2/33730.6 | 31655.9 | **39537.1** |
| ice_hockey | 0.9 | 3.4 | -5.3/3.5 | 33.0 | **79.3** |
| jamesbond | 302.8 | 7869.3 | 440.0/601.5 | 21322.5 | **25354.0** |
| kangaroo | 3035.0 | 10484.5 | 47.0/1632.0 | 1416.0 | **14130.7** |
| krull | 2665.5 | 9930.9 | 9247.6/8147.4 | 11741.4 | **218448.1** |
| kung_fu_master | 22736.3 | 59799.5 | 42259.0/43375.5 | 97829.5 | **233413.3** |
| montezuma_revenge | **4753.3** | 2643.5 | 0.0/0.0 | 2500.0 | 2061.3 |
| ms_pacman | 6951.6 | 2724.3 | 6501.7/7342.3 | 11255.2 | **42281.7** |
| name_this_game | 8049.0 | 9907.1 | 6049.6/21537.2 | 25783.3 | **58182.7** |
| phoenix | 7242.6 | 40092.3 | 33068.2/210996.5 | 224491.1 | **864020.0** |
| pitfall | **6463.7** | -3.5 | -11.1/-1.7 | -0.6 | 0.0 |
| pong | 14.6 | 20.7 | 20.4/**21.0** | 20.9 | **21.0** |
| private_eye | **69571.3** | 15177.1 | 92.4/98.5 | 49.8 | 5322.7 |
| qbert | 13455.0 | 22956.5 | 18901.3/351200.1 | 302391.3 | **408850.0** |
| riverraid | 17118.0 | 16608.3 | 17401.9/29608.0 | **63864.4** | 45632.1 |
| road_runner | 7845.0 | 71168.0 | 37505.0/57121.0 | 222234.5 | **599246.7** |
| robotank | 11.9 | 68.5 | 2.3/13.0 | 73.8 | **100.4** |
| seaquest | 42054.7 | 8425.8 | 1716.9/1753.2 | 392952.3 | **999996.7** |
| skiing | **-4336.9** | -10753.4 | -29975.0/-10180.4 | -10789.9 | -30021.7 |
| solaris | **12326.7** | 2165.2 | 2368.4/2365.0 | 2892.9 | 3787.2 |
| space_invaders | 1668.7 | 2448.6 | 1726.3/43595.8 | **54681.0** | 43223.4 |
| star_gunner | 10250.0 | 70038.0 | 69139.0/200625.0 | 434342.5 | **717344.0** |
| surround | 6.5 | 6.7 | -8.1/7.6 | 7.1 | **9.9** |
| tennis | -8.3 | 23.3 | -1.9/0.5 | **23.9** | -0.1 |
| time_pilot | 5229.1 | 19401.0 | 6617.5/48481.5 | 87085.0 | **445377.3** |
| tutankham | 167.6 | 272.6 | 267.8/292.1 | 272.6 | **395.3** |
| up_n_down | 11693.2 | 64354.3 | 273058.1/332546.8 | 401884.3 | **589226.9** |
| venture | 1187.5 | 1597.5 | 0.0/0.0 | 1773.5 | **1970.7** |
| video_pinball | 17667.9 | 469365.8 | 228642.5/572898.3 | 546197.4 | **999383.2** |
| wizard_of_wor | 4756.5 | 13170.5 | 4203.0/9157.5 | 46204.0 | **144362.7** |
| yars_revenge | 54576.9 | 102759.8 | 80530.1/84231.1 | 148594.8 | **995048.4** |
| zaxxon | 9173.3 | 25215.5 | 1148.5/32935.5 | 42285.5 | **224910.7** |

Table 4: Performance of R2D2 (averaged over 3 seeds) compared to Reactor, IMPALA (shallow and deep expert variants), and Ape-X, with 30 random no-op starts. Ape-X uses best score attained during training, whereas all other agents use final scores.

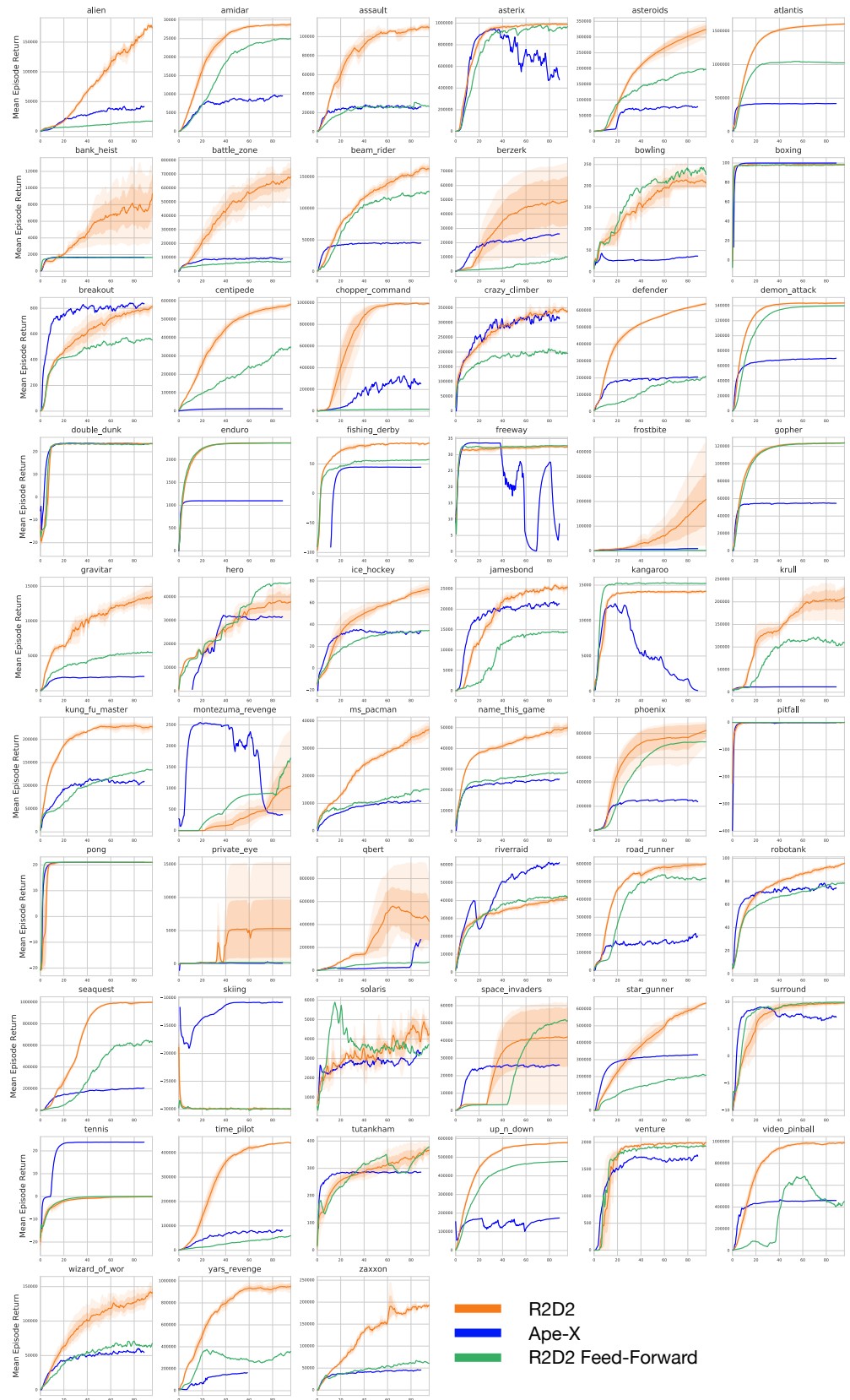

Figure 10: Learning curves on 57 Atari games: R2D2 (3 seeds), Ape-X and R2D2-feed-forward (1 seed each).

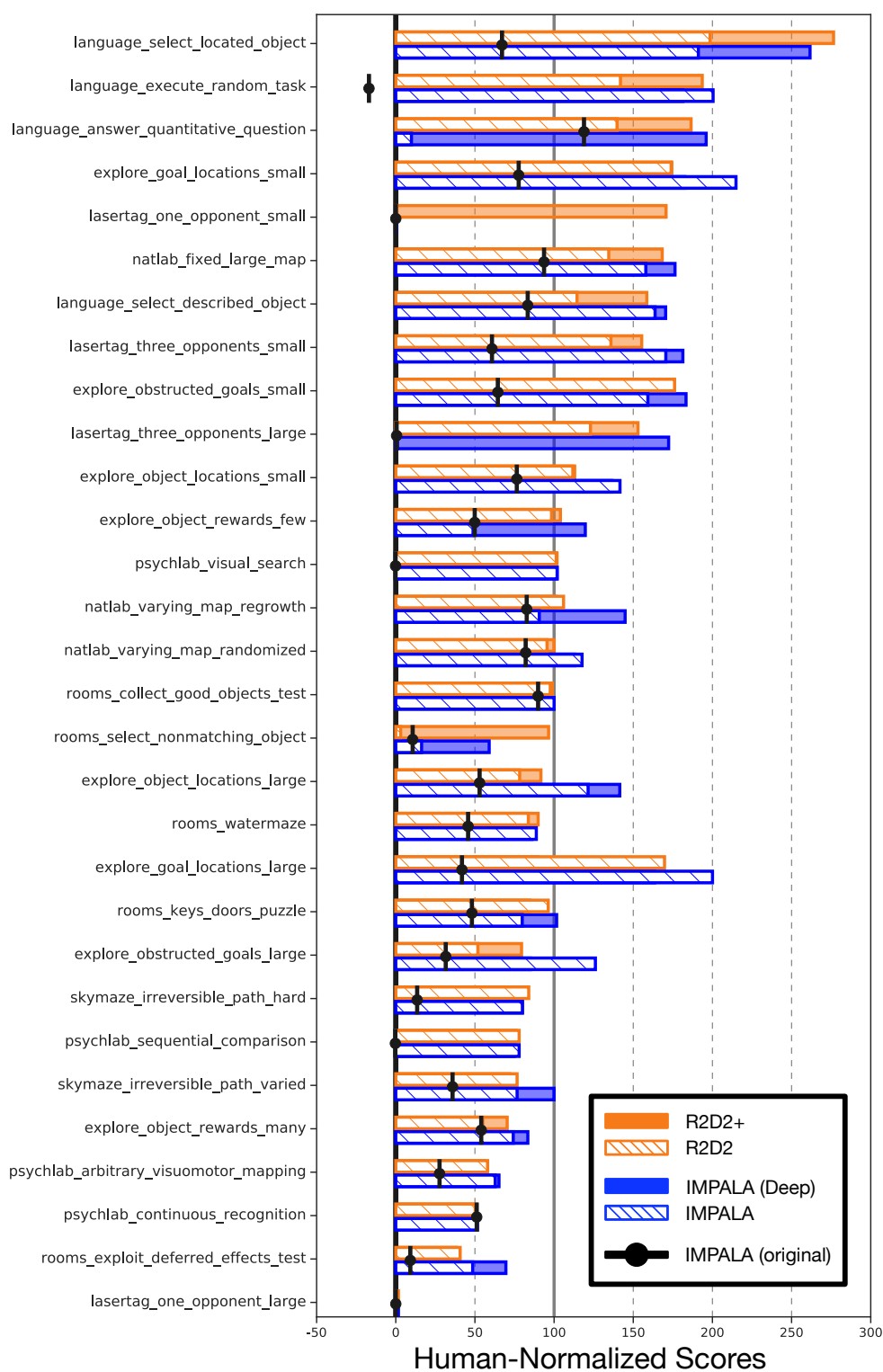

Figure 11: Per-level breakdown of DMLab-30 performance. Comparison between R2D2 and IM-PALA trained for the same number of environment frames. The shallow-network and deep-network versions are overlaid for each algorithm.

