# OpenReview forum: "Recurrent Experience Replay in Distributed Reinforcement Learning"
_ICLR.cc/2019/Conference_

### Official Review · AnonReviewer1 · 2018-11-02
**A thorough investigation of using recurrent networks with experience replay, with impressive results on Atari**

**Rating:** 8
**Confidence:** 4

**Review:**

In this submission, the authors investigate using recurrent networks in distributed DQN with prioritized experience replay on the Atari and DMLab benchmarks. They experiment with several strategies to initialize the recurrent state when processing a sub-sequence sampled from the replay buffer: the best one consists in re-using the initial state computed when the sequence was originally played (even if it may now be outdated) but not doing any network update during the first k steps of the sequence (“burn-in” period). Using this scheme with LSTM units on top of traditional convolutional layers, along with a discount factor gamma = 0.997, leads to a significant improvement on Atari over the previous state-of-the-art, and competitive performance on DMLab.

The proposed technique (dubbed R2D2) is not particularly original (it is essentially “just” using RNNs in Ape-X), but experiments are thorough, investigating several important aspects related to recurrence and memory to validate the approach. These findings are definitely quite relevant to anyone using recurrent networks on RL tasks. The results on Atari are particularly impressive and should be of high interest to researchers working on this benchmark. The fact that the same network architecture and hyper-parameters also work pretty well on DMLab is encouraging w.r.t. the generality of the method.

I do have a couple of important concerns though. The first one is that a few potentially important changes were made to the “traditional” settings typically used on Atari, which makes it difficult to perform a fair comparison to previously published results. Using gamma = 0.997 could by itself provide a significant boost, as hinted by results from “Meta-Gradient Reinforcement Learning” (where increasing gamma improved results significantly compared to the usual 0.99). Other potentially impactful changes are the absence of reward clipping (replaced with a rescaling scheme) and episodes not ending with life loss: I am not sure whether these make the task easier or harder, but they certainly change it to some extent (the “despite this” above 5.1 suggests it would be harder, but this is not shown empirically). Fortunately, this concern is partially alleviated by Section 6 that shows feedforward networks do not perform as well as recurrent ones, but this is only verified on 5 games: a full benchmark comparison would have been more reassuring (as well as running R2D2 with more “standard” Atari settings, even if it would mean using different hyper-parameters on DMLab).

The second important issue I see is that the authors do not seem to plan to share their code to reproduce their results. Given how time consuming and costly it is to run such experiments, and all potentially tricky implementation details (especially when dealing with recurrent networks), making this code available would be tremendously helpful to the research community (particularly since this paper claims a new SOTA on Atari). I am not giving too much weight to this issue in my review score since (unfortunately) the ICLR reviewer guidelines do not explicitly mention code sharing as a criterion, but I strongly hope the authors will consider it.

Besides the above, I have a few additional small questions:
1. “We also found no benefit from using the importance weighting that has been typically applied with prioritized replay”: this is potentially surprising since this could be “wrong”, mathematically speaking. Do you think this is because of the lack of stochasticity in the environments? (I know Atari is deterministic, but I am not sure about DMLab)
2. Fig. 3 (left) shows R2D2 struggling on some DMLab tasks. Do you have any idea why? The caption of Table 3 in the Appendix suggests the absence of specific reward clipping may be an issue for some tasks, but have you tried adding it back? I also wonder if maybe training a unique network per task may make DMLab harder, since IMPALA has shown some transfer learning occurring between DMLab tasks? (although the comparison might be to the “deep experts” version of IMPALA — this is not clear in Fig. 3 — in which case this last question would be irrelevant)
3. In Table 1, where do the IMPALA (PBT) numbers on DMLab come from? Looking at the current arxiv version of their paper, their Fig. 4 shows it goes above 70% in mean capped score, while your Table 1 reports only 61.5%. I also can’t find a median score being reported on DMLab in their paper, did you try to compute it from their Fig. 9? And why don’t you report their results on Atari?
4. Table 4’s caption mentions “30 no-op starts” but you actually used the standard “random starts” setting, right? (not a fixed number of 30 no-ops)

And finally a few minor comments / suggestions:
- In the equation at bottom of p. 2, it seems like theta and theta- (the target network) have been accidentally swapped (at least compared to the traditional double DQN formula)
- At top of p. 3 I guess \bar{delta}_i is the mean of the delta_i’s, but then the index i should be removed
- In Fig. 1 (left) please clarify which training phase these stats are computed on (whole training? beginning / middle / end?)
- p. 4, “the true stored recurrent states at each step”: “true” is a bit misleading here as it can be interpreted as “the states one would obtain by re-processing the whole episode from scratch with the current network” => I would suggest to remove it, or to change it (e.g. “previously”). By the way, I think it would have been interesting to also compare to these states recomputed “from scratch”, since they are the actual ground truth.
- I think you should mention in Table 1’s caption that the PBT IMPALA is a single network trained to solve all tasks
- Typo at bottom of p. 7, “Indeed, Table 1 that even...”

Update: score updated to 8 (from 7) following discussion below

---

> ### Author Response · Authors · 2018-11-20
> **RE: A thorough investigation...**
>
> Thank you for your comments and in particular your concerns around the use of importance weighting. We took your concerns to heart and have (as we discuss below) included it and rerun the experiments.
>
> “The fact that the same network architecture and hyper-parameters also work pretty well on DMLab is encouraging w.r.t. the generality of the method.”
>
> We want to thank the reviewer for making note of this aspect. It is something we consider particularly noteworthy considering common problems with robustness in deep RL.
>
> “… a couple of important concerns though. The first one is that a few potentially important changes were made to the “traditional” settings typically used on Atari, which makes it difficult to perform a fair comparison to previously published results.”
>
> This is a very reasonable concern and we have run a more thorough set of ablations on R2D2 which have now been included in the latest revision. These are not 100% completed yet, but are far enough along to give a clear picture. Specifically, we are taking your recommendations and comparing R2D2 with (1) Feed-forward only (already included, but now also done over all 57 Atari games), (2) Reward clipping but no value function rescaling, (3) Smaller discount (gamma = 0.99), and (4) end-episode on life-loss enabled. Only the last one is not included in the current revision, but will be included before revisions close.
>
> “The second important issue I see is that the authors do not seem to plan to share their code to reproduce their results. Given how time consuming and costly it is to run such experiments, and all potentially tricky implementation details (especially when dealing with recurrent networks), making this code available would be tremendously helpful to the research community (particularly since this paper claims a new SOTA on Atari)...  I strongly hope the authors will consider it.”
>
> Distributed training, in particular, is an area where publically available open source code goes a long way towards reproducibility and progress in the field. Although we are not able to immediately release the code, we believe that we will be able to make the source available in the future.
>
>
> “1. “We also found no benefit from using the importance weighting that has been typically applied with prioritized replay”: this is potentially surprising since this could be “wrong”, mathematically speaking. Do you think this is because of the lack of stochasticity in the environments? (I know Atari is deterministic, but I am not sure about DMLab)”
>
> Thank you for pointing this out! This does make sense and we agree that in principle the lack of importance weighting when using prioritized replay is not well supported. We have now included it in the algorithm and rerun almost all of our experiments (ablations are still in progress). We have not found it necessary to retune hyper-parameters to support this change, and in fact found that re-introducing importance weighting did stabilise training on some of the DMLab levels and slightly improved overall performance.
>
> “2. Fig. 3 (left) shows R2D2 struggling on some DMLab tasks. Do you have any idea why?“
>
> We do not use the asymmetric reward clipping of IMPALA and believe that this clipping is most helpful in some of the language tasks. We are in the process of running a very small test in which we add the asymmetric clipping to verify, but do not plan to add it to the algorithm itself in the interest of generality.
>
> Additionally, one of the larger benefits of IMPALA PBT is the use of population-based training, and we suspect this is another reason for IMPALA occasionally out-performing R2D2.
>
> “3. In Table 1, where do the IMPALA (PBT) numbers on DMLab come from?”
>
> We obtained the IMPALA results data from the authors of the cited paper: “Multi-task Deep Reinforcement Learning with PopArt”. However, after further discussion we believe the best approach would be to rerun IMPALA on our same hardware and training regime. Until this completes we will use the provided data, but again, we hope to replace this before the final revision.
>
> “And finally a few minor comments / suggestions:”
>
> Thank you for these comments and suggestions, we have made the corresponding edits to clarify things and fix mistakes. We should also mention that while doing this we fixed a bug that limited Atari training episode times to 14 minutes (50K frames) instead of 30 minutes (108K frames), this slightly improves some of our Atari results.

---

> > ### Comment · AnonReviewer1 · 2018-11-23
> > **RE: A thorough investigation...**
> >
> > Thanks for the reply and significant additional effort in the revision. I'm going to update my score to 8 to reflect this -- I'd actually be willing to increase it further if full experimental results were available already.
> >
> > Regarding experimental results, I very much appreciate the (soon-to-be-)full training curves on all 57 Atari games, but please also add the aggregated human-normalized median score (having it added to Fig. 8 for all ablation experiments would be ideal). By the way, in Fig. 8, having a lower sample efficiency than Rainbow / Reactor is indeed expected since these have much fewer actors (and thus collect data much slower), however the difference with Ape-X is a bit more surprising. Is it because the RNN is slower to train? Or maybe the higher discount factor? (another reason to add ablations to this plot!)
> >
> > "We obtained the IMPALA results data from the authors of the cited paper": in that case please mention it in the paper, since some of these results look worse than the published ones (I'm referring to previous comment "Looking at the current arxiv version of their paper [https://arxiv.org/pdf/1809.04474.pdf], their Fig. 4 shows it goes above 70% in mean capped score, while your Table 1 reports only 61.5%" ==> this may deserve being validated with the authors)
> >
> > I didn't have time to carefully re-read the revised paper, but I noticed a small typo in Fig. 1(a) where t+1 should be t+m.
> >
> > I also really hope you manage to open source (at least part of) your code in the near future.
> >
> > Thanks!

---

> > > ### Author Response · Authors · 2018-11-23
> > > **RE: A thorough investigation...**
> > >
> > > Thank you very much for the repeated careful reading of the paper, and thank you for kindly improving your score!
> > >
> > > Following your suggestion, we have added full Atari-57 results including the respective median human-normalized score for the feed-forward ablation (Table 1). For the other ablations (discount, life loss signal, value function rescaling vs clipped rewards) we considered a full Atari-57 ablation to be excessively costly in terms of computational resources and were not intending to provide those (beyond the already included 5 games x 3 seeds each). We could run those for the final version of the paper, with a single seed each, if you consider this data to be especially valuable, but would otherwise opt to use the chosen 5 games as representative.
> > >
> > > Your observation about the lower sample efficiency compared to Ape-X, this is indeed a great remark, and we have added a paragraph in the appendix to address this. Because of the very different batch characteristics of our sequence-based replay (64x80 instead of 512x1), our learner runs at approximately 1/4 of the updates-per-second compared to Ape-X, which results in a smaller ‘replay ratio’ (expected number of times an experienced observation is being replayed): approximately 0.8 (R2D2) vs 1.3 (Ape-X). We believe this mostly explains the initial difference in sample efficiency that you pointed out.
> > >
> > > As suggested, we included a footnote mentioning the results we obtained in personal communication with the IMPALA-PopArt paper authors. Regarding the performance difference you are pointing out, we note that we are only comparing to the IMPALA, not the PopArt-IMPALA reported in that publication (solid blue curve in their Fig. 4), as this seemed to be the most apples-to-apples comparison.
> > >
> > > We have just uploaded a new revision with the above changes - a final revision including the life-loss ablation will be uploaded before the deadline.

---

> > > > ### Comment · AnonReviewer1 · 2018-11-26
> > > > **RE: A thorough investigation...**
> > > >
> > > > Thanks for the clarifications. Sorry, I had misunderstood your intent regarding ablations. I do realize that a full ablation would be quite time-consuming and costly, personally I'm fine without them -- though of course if you're willing to add more results I won't object ;)
> > > >
> > > > There's one small thing I think you should be able to do easily, which would be to add "R2D2, Feed-Forward" to Fig. 8.

---

### Official Review · AnonReviewer3 · 2018-11-02
**The proposed RL agent leads to interesting results but the technical details need to be clarified**

**Rating:** 7
**Confidence:** 2

**Review:**

Summary:
Leveraging on recent advances on distributed training of RL agents, the paper proposes the analysis of RNN-based RL agents with experience replay (i.e., integrating the time dependencies through RNN). Precisely, the authors empirically compare a state-of-the-art training strategy (called zero start state) with three proposed training strategies (namely; zero-state with burn-in, stored-state and stored-state with burn-in). By comparing these different strategies through a proposed metric (Q-value discrepancy), the authors conclude on the effectiveness of the stored-state with burn-in strategy which they consider for the training of their proposed Recurrent Replay Distributed DQN (R2D2) agent.

The proposed analysis is well-motivated and has lead to significant results w.r.t. the state-of-the-art performances of RL agents.

Major concerns: My major concerns are three-fold:
- The authors do not provide enough details about some "informal" experiments which are sometimes important to convince the reader about the relevance of the suggested insights (e.g., line 3 page 5). Beyond this point, the paper is generally hard to follow and reorganizing some sections (e.g., sec. 2.3 should appear after sec. 3 as it contains a lot of technical details) would certainly make the reading of the paper easier.
- Hausknecht & Stone (2015) have proposed two training strategies (zero-state and Replaying whole episode trajectories see sec. 3 page 3). The authors should clarify why they did not considered the other states in their study.
- The authors present results (mainly, fig. 2 and fig. 3) suggesting that the proposed R2D2 agent outperform the agents Ape-X and IMPALA, where R2D2 is trained using the aforementioned stored-state with burn-in strategy. It is not clear which are the considered training strategies adopted for the (compared to) state-of-the-art agents (Ape-X and IMPALA). The authors should clarify more precisely this point.

Minor concerns:
- The authors compare the different strategies only in terms of their proposed Q-value discrepancy metric. It could be interesting to consider other metrics in order to evaluate the ability of the methods on common aspects.

---

> ### Author Response · Authors · 2018-11-20
> **RE: The proposed RL agent...**
>
> Thank you for raising these concerns. We have attempted to address them in this revision.
>
> “The authors do not provide enough details about some "informal" experiments...”
>
> We have now significantly revised our LSTM training analysis to include a more detailed study that shows representation drift measured by both parameter lag (number of updates since experience was generated) and the q-value discrepancy, and for the same runs the mean episodic return, some of which is contained in the appendix. Additionally, we now have results as we vary burn-in from zero to 20 and up to 40 steps. We think this improves the section quite a bit, but we are still looking at edits to the paper to improve clarity further.
>
> “Beyond this point, the paper is generally hard to follow and reorganizing some sections (e.g., sec. 2.3 should appear after sec. 3 as it contains a lot of technical details)...”
>
> We have moved one of these sections to the appendix and tried to improve the flow of the paper. Please let us know if this makes for a clearer read.
>
> “Hausknecht & Stone (2015) have proposed two training strategies (zero-state and Replaying whole episode trajectories see sec. 3 page 3). The authors should clarify…”
>
> Hausknecht and Stone (2015) argued that the two strategies performed similarly in their experiments. We agree that a more thorough investigation of whole-trajectory-based training would be valuable, with special attention to sample correlation, variance and optimization settings. However, this seems to exceed the scope of the paper.
>
> “The authors present results (mainly, fig. 2 and fig. 3) suggesting that the proposed R2D2 agent outperform the agents Ape-X and IMPALA, where R2D2 is trained using the aforementioned stored-state with burn-in strategy. It is not clear…”
>
> In all R2D2 experiments outside of the initial analysis section (where we specifically study these methods) we used the stored-state with 40-step burn-in method. We have attempted to make this more explicit in the revision. Ape-X does not use an RNN, and IMPALA is perhaps most like the "entire episode trajectories" approach in Hausknecht and Stone, but due to using the mostly on-policy actor-critic is hard to compare directly. To avoid any confusion, the reported Ape-X and IMPALA results are not our own reruns of the algorithms, but taken from their respective publications or private communication with the authors.
>
> “The authors compare the different strategies only in terms of their proposed Q-value discrepancy metric. It could be interesting to consider other metrics in order to evaluate the ability of the methods on common aspects.”
>
> As mentioned above, we have attempted to significantly improve this by including more information.

---

> > ### Comment · AnonReviewer3 · 2018-11-26
> > **Reviewer 3 additional comments**
> >
> > We thank the authors for the provided additional details. After reading their responses and other reviewers comments, I upgrade my rating to 7.

---

### Official Review · AnonReviewer2 · 2018-11-05
**Rrecurrent NNs in distributted RL settings as a clear improvement of the feed-forward NN variations in partially observed environments**

**Rating:** 7
**Confidence:** 3

**Review:**

This paper investigates the use of recurrent NNs in distributted RL settings as a clear improvement of the feed-forward NN variations in partially observed environments. The authors present "R2DR" algorithm as a A+B approach from previous works (actually, R2D2 is an Ape-X-like agent using LTSM), as well as an empirical study of a number of ways for training RNN from replay in terms of the effects of parameter lag (and potential alleviating actions) and sample-afficiency. The results presented show impressive performance in Atari-57 and DMLab-30 benchmarks.

In summary, this is a very nice paper in which the authors attack a challenging task and empirically confirm that RNN agents generalise far better when scaling up through parallelisation and distributed training allows them to benefit from huge experience. The results obtained in ALE and DMLab improves significantly upon the SOTA works, showing that the trend-line in those benchmarks seem to have been broken.

Furthermore, the paper presents their approach/analyses in a well-structured manner and sufficient clarity to retrace the essential contribution. The background and results are well-contextualised with relevant related work.

My only major comments are that I’m a bit skeptical about the lack of a more thorough (theoretical) analysis supporting their empirical findings (what gives me food for thought is that LSTM helps that much on even fully observable games such as Ms. Pacman); and the usual caveats regarding evaluation: evaluation conditions aren't well standardized so the different systems (Ape-X, IMPALA, Reactor, Rainbow, AC3 Gorilla, C51, etc.) aren't all comparable. These sort of papers would benefit from a more formal/comprehensive evaluation by means of an explicit enumeration of all the dimensions relevant for their analysis: the data, the knowledge, the software, the hardware, manipulation, computation and, of course, performance, etc. However only some of then are (partially) provided.

---

> ### Author Response · Authors · 2018-11-20
> **RE: Recurrent NNs in distributed RL...**
>
> Thank you for your comments and suggestions.
>
> “... only major comments are that I’m a bit skeptical about the lack of a more thorough (theoretical) analysis supporting their empirical findings (what gives me food for thought is that LSTM helps that much on even fully observable games such as Ms. Pacman);”
>
> We do not have theoretical contributions to add in our rebuttal, but would like to offer some additional resources for understanding the empirical findings. In addition to the more thorough reporting of results now included, as well as now including 3 seeds in all R2D2-based experiments, we have uploaded videos of the agent’s learned policy on a handful of Atari games. What we observe in these videos is that R2D2 has learned to leverage memory in Atari in unexpected ways. That is, Atari *does* directly benefit from long-term memory in several specific cases. For example, in MsPacman the agent learns to precisely time the ghosts’ vulnerability in order to obtain well timed multi-ghost-meals, yielding much higher scores.
>
> Please note these videos are uploaded anonymously to youtube and marked unlisted, which should prevent us from inferring any geographic information about reviewers who view them.
>
> MsPacman
> R2D2: https://youtu.be/eexCo9wHqfU
> R2D2 Feed-Forward: https://youtu.be/stI08CZlKqo
>
> SeaQuest
> R2D2: https://youtu.be/5Umrkdis8OY
> R2D2 Feed-Forward: https://youtu.be/8o1LcK_3S3U
>
> QBert
> R2D2: https://youtu.be/UUn_vXj89Ps
> R2D2 Feed-Forward: https://youtu.be/UUn_vXj89Ps
>
>
> “and the usual caveats regarding evaluation…”
>
> To this end we have run additional ablations on our architectural choices and are in the process of rerunning IMPALA on the same hardware and training regime as we used for R2D2. We have also attempted to clarify our exact evaluation regime by pointing out the 30-minute episode timeout on Atari, the fact that our results are final scores (not max-over-training as have been reported for e.g. Ape-X), and other evaluation details.

---

### Public Comment · (anonymous) · 2018-09-30
**Questions on evaluation conditions and sample efficiency**

Congratulations on these very impressive results! I have a few comments and questions on evaluation conditions and sample efficiency:

1.  Regarding no-op vs. human starts: Table 1 seems to use median scores from Ape-X in the human start condition (358.1%, vs. the 358% listed in Table 1 of the Ape-X paper). But only no-op scores are listed in the Appendix of this paper, and it also seems plausible that you are referring to the 20 hr no-op score of Ape-X.  Could you clarify what evaluation condition you're using in this paper, and which condition you are comparing against in Ape-X?

2.Could you clarify how many frames are used per game by R2D2? A naive conversion from Ape-X to R2D2, using Table 1 from the Ape-X paper, accounting for the reduced wall time (3 vs 5 days) and number of actors (360 vs. 256) seems to suggest roughly 10 billion frames per game, but I'm not sure if I'm missing something that would affect these calculations. Also, it would be interesting to know just how severe the remaining sample efficiency gap mentioned in the conclusion is - median scores for Rainbow vs. R2D2 with comparable frames might be illustrative, for example.

---

> ### Author Response · Authors · 2018-10-01
> **Re: Questions on evaluation conditions and sample efficiency**
>
> 1. Thank you for catching this, we inadvertently included the human-start values for Ape-X instead of the no-op starts evaluation results. All mean/median values throughout the paper are referring to 30 no-op evaluation conditions. The correct values for Ape-X should be mean=434.1%,  median=1695.6%. The other values in the table correctly refer to 30 no-op results for the other agents.
>
> 2. On Atari, the R2D2 agent consumes approximately 5.7B environment frames per day (with 256 actors, each running at approximately 260 fps), i.e. approximately 17B for the reported 3-day version. On DMLab, the frame rate is around half of the Atari frame rate, ~2.65B env frames per day, i.e. ~10.6B frames for the entire 4-day run whose results we report. Regarding a comparison with Rainbow in terms of environment frames, we agree this is informative and will include such a comparison when revisions open up.

---

> > ### Public Comment · (anonymous) · 2018-10-04
> > **Thanks for clarifying**
> >
> > Thanks for the clarification on frames + no-op/human starts! I look forward to the Rainbow comparison, as well. P.S. I assume you meant median=434.1, mean=1659.6 :)

---

### Public Comment · (anonymous) · 2018-10-01
**Minor contribution**

The paper proposes to replace the feed-forward NN with recurrent version in Distributed RL setting, solving few technical issues (such as how to properly initialize the RNN), and shows improved performance in Atari-57 and DMLab-30 domain.

While the experimental results are impressive, the contribution of the paper is rather straightforward application of the RNN. I'd consider it only a minor improvement over the state-of-the art and would like to see either
a) more thorough analysis why it improves performance in almost-observable MDPs, or
b) more fundamental additional contribution
to consider it for publication.

---

> ### Author Response · Authors · 2018-10-02
> **Our contributions**
>
> Thank you for your interest in our paper!
>
> In fact, our work started out as an investigation of a novel algorithmic technique.
> We developed an Ape-X-like agent enhanced with an LSTM as a platform for this research.
>
> The empirical improvement over previous agents was sufficiently surprising that we considered it worth further study, isolated from confounding factors from unrelated techniques. Towards the aim of understanding R2D2's empirical performance, we contributed a study of the effects of parameter lag on representation drift and recurrent state staleness, potential mitigation strategies, and the memory-dependence in a learned recurrent policy.
>
> This paper focuses on presenting and analyzing effective ways of training a recurrent memory from replay, resulting in a more universal agent achieving SOTA results across multiple environment suites with a single set of parameters. While more analysis remains to be done to fully understand the role of memory in RL, we believe the above contributions provide valuable insights to the RL community.

---

### Public Comment · (anonymous) · 2018-10-01
**Parameter notation mixup**

I think you have a mixup in the target/online parameter notation at the bottom of page 2. The argmax should be computed using the online parameters and the target value computed using the target parameters.

---

> ### Author Response · Authors · 2018-10-02
> **Re: Parameter notation**
>
> Thank you for your interest in our paper, and thank you for catching this bug, we will fix this in a future revision.

---

### Author Response · Authors · 2018-11-20
**All Reviewers**

We first want to thank all the reviewers and commenters for their close reading and constructive feedback. We have revised or expanded most of our experiments and attempted to clarify the text in various locations. Additionally, as requested we are including (in the appendix) a figure comparing the sample efficiency of R2D2 with Rainbow, Ape-X, and Reactor on Atari. As detailed in our individual responses we have extended the ablations and rerun our experiments to address some concerns. This resulted in slightly improved performance, which is now averaged over three seeds.

We intend to add further ablation results (on life-loss signal) and our own rerun of IMPALA with matched action-set before the revision period ends.

---

> ### Author Response · Authors · 2018-11-27
> **All Reviewers**
>
> We thank all reviewers once again for the careful reading of the paper and the helpful comments.
>
> We have updated our ablations, now including two life-loss ablations, and provided complete details on the feed-forward ablation including human-normalized scores and sample efficiency data.
>
> Finally, we have updated the paper with our own re-run of the IMPALA agent on DMLab with the new action-set and longer training time, for a fairer comparison with R2D2. To explore the potential of our agent further, we also added a version of R2D2 more closely matching the Deep IMPALA architecture (deep ResNet + asymmetric reward clipping). Both our re-run of IMPALA and R2D2 achieve new SOTA scores on DMLab-30. We intend to report all DMLab-30 scores at 10B environment frames, but have restricted to 5B frames for this revision as these runs have not all completed at the time of the revision deadline.

---

> > ### Comment · AnonReviewer1 · 2018-11-27
> > **Re: All Reviewers**
> >
> > Thanks, very interesting! Could you please just add a short description of R2D2+ in Table 1's caption, for the final version?

---

> > > ### Author Response · Authors · 2018-11-29
> > > **Re: All Reviewers**
> > >
> > > Absolutely, thanks!

---

### Meta-Review · Area_Chair1 · 2018-12-13
**Valuable insights on training reinforcement learning with recurrent neural networks at scale**

**Confidence:** 4
**Recommendation:** Accept (Poster)

**Metareview:**

The paper proposes a new distributed DQN algorithm that combines recurrent neural networks with distributed prioritized replay memory. The authors systematically compare three types of initialization strategies for training the recurrent models. The thorough investigation is cited as a valuable contribution by all reviewers, with reviewer 1 noting that the study would be of interest to "anyone using recurrent networks on RL tasks". Empirical results on Atari and DMLab are impressive.

The reviewers noted several weaknesses in their original reviews. These included issues of clarity, a need for more detailed ablation studies, and need to more carefully document the empirical setup. A further question was raised on whether the empirical results could be complemented with theoretical or conceptual insights.

The authors carefully addressed all concerns raised during the reviewing and rebuttal period. They took exceptional care to clarify their writing, document experiment details, and ran a large set of additional experiments as suggested by the reviewers. The AC feels that the review period for the paper was particularly productive and would like to thank the reviewers and authors.

The reviewers and AC agree that the paper makes a significant contribution to the field and should be accepted.